# Recognition Method of Massage Techniques Based on Attention Mechanism and Convolutional Long Short-Term Memory Neural Network

**DOI:** 10.3390/s22155632

**Published:** 2022-07-28

**Authors:** Shengding Zhu, Jingtao Lei, Dongdong Chen

**Affiliations:** School of Mechatronic Engineering and Automation, Shanghai University, Shanghai 200444, China; shaunzhu@shu.edu.cn (S.Z.); jtlei2000@163.com (J.L.)

**Keywords:** tactile sensor, variational auto-encoder, recurrent neural network, attention mechanism

## Abstract

Identifying the massage techniques of the masseuse is a prerequisite for guiding robotic massage. It is difficult to recognize multiple consecutive massage maps with a time series for current human action recognition algorithms. To solve the problem, a method combining a convolutional neural network, long-term neural network, and attention mechanism is proposed to identify the massage techniques in this paper. First, the pressure distribution massage map is collected by a massage glove, and the data are enhanced by the conditional variational auto-encoder. Then, the features of the massage map group in the spatial domain and timing domain are extracted through the convolutional neural network and the long- and short-term memory neural network, respectively. The attention mechanism is introduced into the neural network, giving each massage map a different weight value to enhance the network extraction of data features. Finally, the massage haptic dataset is collected by a massage data acquisition system. The experimental results show that a classification accuracy of 100% is achieved. The results demonstrate that the proposed method could identify sequential massage maps, improve the network overfitting phenomenon, and enhance the network generalization ability effectively.

## 1. Introduction

The fast pace of life and high work pressure lead to common sub-health phenomena, such as backache and lack of vitality. Massage is a physical naturopathic therapy that relieves fatigue and pain and improves sub-health problems [1]. At present, the demand for masseuses at home and abroad is strong, but the gap is large, and the technique level is uneven. Robotic massage will replace human massage services with the development of robot technologies. To realize robotic massage, it is crucial to systematically understand the massage techniques of professional masseurs, explore the features of massage techniques, and provide a reference for robots to reproduce the massage techniques of masseurs.

A tactile sensor is required to obtain massage technique information about a masseuse; it is a sensor used to imitate the tactile function of animals and humans. It can sense the force between the sensor and the contact object, the shape and temperature of the detected object, etc. According to the working principle, tactile sensors can be divided into piezoresistive, capacitive, piezoelectric, inductive, photoelectric, etc. Currently, the first three types of tactile sensors are commonly used. Tactile sensors have been widely used in many fields, such as teaching [2,3], medical care [4], virtual reality [5] and other fields. Literature [6] proposed a capacitive tactile sensor for collecting hand information when manipulating clothing, but the sensor has a small range and few touch-sensing units, so it can’t reflect the detailed information of the operator’s hand. In the literature [7], a capacitive tactile sensor is proposed. Its electrodes are made using flexible circuit board technology. Air is used as the sensor dielectric material, and a layer of silica gel spacer is added in the middle of the sensor to increase the range of deflection of the electrode layer. This kind of sensor has a simple manufacturing process, but its range is small, and the signal acquisition circuit is complicated. Literature [8] proposed a fabric piezoresistive flexible tactile sensor, which has high flexibility and excellent ductility, and the acquisition circuit can be automatically disconnected when the sensor is not under force to reduce heat generation, but its single sensing is too large to obtain detailed information about human hands. Most of the current tactile sensors are not sufficiently flexible, and the distribution of their sensing units is not dense enough, so it is difficult to obtain detailed massage information about the massager’s hand. Most capacitive tactile sensors have small measuring ranges and complex signal acquisition circuits. They are difficult to manufacture and are easily disturbed by wet environments; most piezoelectric tactile sensors have few sensing units and serious temperature drifts. Therefore, based on the piezoresistive principle and flexible circuit board technology, this paper designs a flexible tactile sensor with a large range and high robustness, and integrates it in the palm and five fingers of the fabric glove to collect tactile information on the massager’s hand.

To guide the robot massage, it is far from enough to obtain only the massage technique information. It is also necessary to extract the features of the massage techniques through algorithms to realize the purpose of technique identification, and so on. The artificial intelligence recognition algorithm can be roughly divided into two categories: a classical machine learning algorithm and a deep learning algorithm using a neural network. With the continuous development of artificial intelligence technology, recognition methods based on machine learning and deep learning are widely used in wearable devices, unmanned driving, intelligent medical treatment, and other fields.

Some scholars have used machine learning to establish computational models for recognition or prediction [9,10,11]. Commonly used machine learning recognition algorithms are Decision Tree [12,13], K-Nearest Neighbor [14,15], Hidden Markov Model [16,17] and Support Vector Machine [18,19]. Reference [20] uses a variety of sensors to collect human data, performs principal component analysis on the data collected by the sensors, extracts 10 main features, and uses the multi-SVM algorithm to identify 35 human behaviors, with an average recognition accuracy of 86%. Reference [21] placed a gyroscope on the subject’s foot, collected angular velocity data, and completed the identification of behaviors such as walking, running, and going up and down stairs based on a Decision Tree algorithm. Reference [22] proposed a real-time time series segmentation method, aiming at six human behaviors, using principal component analysis and the K-nearest neighbor algorithm to complete behavior identification. However, classical machine learning algorithms all require manual extraction of data features, and processing of features such as selection and dimensionality reduction, which is not only difficult to extract high-level abstract features of data, but also laborious.

Deep learning algorithms don’t require manual feature extraction. The network can automatically extract high-level abstract features of data based on back-propagation algorithms. The recognition accuracy of deep learning algorithms is generally higher than that of classical machine learning algorithms. With in-depth research on deep learning algorithms and the improvement of computer performance, deep learning methods have made continuous breakthroughs in the fields of image and speech recognition. Reference [23] uses a 3D convolution kernel to extract features in the data space domain as well as in the data timing domain, obtain the motion information of multiple consecutive frame data, and regularize the output results. Good recognition accuracy has been achieved. Reference [24] combines the convolutional neural network CNN and the long short-term memory neural network LSTM, uses the convolutional neural network to extract the spatial domain features of the data, and gives it to the long short-term memory network as an input to extract the timing domain information of the data. The network achieves 73.1% and 88.6% recognition accuracy on the Sports 1 million and UCF-101 datasets, respectively. Reference [25] made a self-made dataset containing 12 different static gesture pictures, and trained on this dataset based on convolutional neural network CNN. Experiments show that compared with other methods, CNN is more sensitive to static gestures in complex environments, but recognition accuracy is low for gestures with rotation. Reference [26] proposed a system for human behavior recognition based on wearable smart devices, using a two-dimensional convolutional neural network CNN to recognize 9 human behaviors in a home environment, and achieved good recognition results. Reference [27] developed a tactile glove that can collect normal force. The volunteer wears the tactile glove and grabs the object to obtain a visual image of the human hand’s normal force and uses the Resnet-18 structure network to recognize 6 kinds of grasping postures. A recognition accuracy of 89.4% is obtained. Reference [28] fuses the convolutional layer and the long-short-term memory unit to the long-term memory convolution layer. This layer and the standard long-short-term memory unit are used in series to extract the spatial and timing features of the acceleration sensor. The experimental results show that this method achieves higher accuracy with a shorter sliding window for data acquisition. However, the recognition of time series-related data in deep learning is mainly for speech recognition, and the recognition of images is mainly for single-frame images. A massage technique often corresponds to multiple frames of images, so it is necessary to design a network structure that can extract the timing features of sequential massage maps. Therefore, a 2D convolutional neural network and a long-short-term memory neural network are combined. The long-short-term memory neural network used for speech feature extraction is applied to extract the timing features of massage map groups. The channel attention mechanism is improved to the frame attention mechanism, and it is introduced between the convolutional neural network and the recurrent neural network. The convolution neural network is used to automatically extract the deep abstract spatial domain features of each frame of the massage maps, and the long- and short-term memory network is used to extract the timing features between frames in each group of tactile data. Finally, the sensor data are sent to the Soft Max classifier to realize the recognition of the massage techniques. In addition, using traditional methods to make a massage dataset takes a long period and a lot of effort, so it is necessary to use an algorithm to expand the dataset.

In view of the above problems, this paper conducts research on massage manipulation recognition based on flexible tactile sensors and deep learning. The tactile sensor integrated on the glove is used to collect massage information about the masseuse, the sensor data are preprocessed and expanded, and a neural network is constructed to realize the classification of massage maps with time series. The contributions of this paper are as follows:(1)Design and manufacture a dot-matrix flexible tactile sensor that can be integrated on gloves and its data acquisition system to collect detailed massage information of the masseuse’s hand; realize the key frame extraction of the massage map group based on the frame difference method; and use the conditional variational auto-encoder to expand the massage data of the key frame. Thus, the preprocessing of the massage data and the expansion of the dataset are completed;(2)Realize the key frame extraction of the massage map group based on the frame difference method and use the conditional variational auto-encoder to expand the massage data of the key frame. Thus, the preprocessing of the massage data and the expansion of the dataset are completed;(3)Combine the convolutional network and the recurrent neural network to establish a neural network structure that can realize the training of sequential massage map groups. The channel attention mechanism is improved into the frame attention mechanism and introduced into the network to achieve autonomous training of the weights of each frame of tactile data. Thus, the ability to extract features from network data are enhanced. Then, build a massage technique identification experimental system, carry out massage technique identification test research, extract massage technique features, and provide a solid foundation for guiding robotic massage applications.

The remainder of this paper is organized as follows. Section 2 proposes the massage data acquisition system and the massage haptic dataset. The neural network structure for massage technique recognition is presented in Section 3. Section 4 describes the experiment and the results. The conclusions are presented in Section 5.

## 2. Massage Data Acquisition System

### 2.1. Tactile Sensor and Data Acquisition System

Circular copper sheets with a diameter of 5 mm, as well as row and column electrodes, are deposited on the polyimide substrate by the flexible circuit board process, and electrodes are drawn out with a flexible circuit board connector to make the upper and lower electrode layers of the flexible sensor. The conductive double-sided tapes are pasted on the Velostat material (developed by Custom Materials) as a piezoresistive material. Then, the piezoresistive material is cut out with a punch into a number of rounds with a diameter of 6 mm. They were pasted on the circular copper sheet of the lower electrode sheet. The upper electrode sheet and the lower electrode sheet are integrated with double-sided tape to fabricate a flexible tactile sensor with a sandwich structure. In this way, sensors with sensor units of 8 × 8 and 2 × 11 are fabricated and integrated into the palm of the fabric glove and the first knuckle region of the five fingers by sewing to make massage gloves. The physical map of the massage gloves is shown in Figure 1a.

Figure 1b shows the working principle of the tactile sensor. The middle layer of the tactile sensor is a piezoresistive material, and its magnitude of resistance varies with the magnitude of the normal force applied to the sensing unit. The sensor is connected to the voltage divider circuit through the multiplexer, and the multiplexer controls the sensing unit to be connected to the voltage divider circuit. Voltage of sensing units is collected through the digital-to-analog conversion function of the lower computer to obtain resistance of the middle layer of sensing units, and the resistance is obtained by the following formula:(1)$$R_{sensor}=\frac{V_{sensor}\cdot R_{fixed}}{Vcc-V_{sensor}}\;$$
where *R_sensor_* is the resistance of piezoresistive material of a single sensing unit, *R_fixed_* is the resistance of the divider resistor in the voltage divider circuit, *V_CC_* is the source voltage of the voltage divider circuit, and *V_sensor_* is the voltage applied on the sensing unit, which is selected by the multiplexer.

Figure 2 is a block diagram of the massage data acquisition system. The flexible tactile sensor integrated on the glove is connected to the multiplexer through the FPC connectors and the FPC cables, and then connected to the voltage divider circuit. The main control chip is the STM32, which controls the on-off of the multiplexer and combines the ADC function to scan the voltage of each sensing unit one by one and send it to the upper computer. Matlab visualizes the force of each sensing unit of the sensor. The force of the sensing unit from small to large corresponds to the color from cold to warm, and finally, the pressure distribution map of the tactile sensor is obtained.

### 2.2. Massage Haptic Dataset

After completing the design of the massage data acquisition system, it is necessary to first collect a pressure distribution map of the massage techniques. Second, the key frames of the massage map groups are extracted by the frame difference method, and the redundant frames are discarded. Finally, a neural network is built based on the principle of the conditional variational auto-encoder to expand the massage map groups to complete the production of the dataset.

#### 2.2.1. Massage Data Collection

Three types of massage techniques are designed: clockwise rubbing against the human scapula with the palm, counterclockwise rubbing with the palm against the human scapula, pressing with the five fingers in turn on the human scapula, and rubbing the scapula back and forth once with the palm.

The ADC clock frequency of the lower computer of the data acquisition system is configured as 9 MHz, and the sampling period is 55 cycles. The lower computer is connected to the upper computer through the uart, the upper computer visualizes and saves the collected voltage data of sensing units, and the sampling frequency of the visualized tactile frames is about 12 Hz.

Two volunteers were invited to collect massage data. One of the volunteers wore the designed massage glove on the right hand and massaged on the other volunteer’s scapula according to the designed three types of massage techniques to collect data. Each type of massage technique was repeated 100 times, and 100 sets of data were collected for each type of massage technique. Each set of data contained 60 frames of visualized massage maps. The massage data collection process diagram is shown in Figure 3.

#### 2.2.2. Massage Data Process

The massage data collection system is used to collect data for the above three massage techniques. If the sensor collection frequency is fast, there may be many redundant frames in the initially collected frames. If it is sent directly to the neural network, the training parameters will be greatly increased. If it increases, it may cause network overfitting and reduce recognition accuracy. Therefore, the frame difference method is first used to get the pixel difference between every two adjacent frames of the massage map group, then extract the 20 frames with the largest difference between adjacent pixels in each massage map group as key frames.

In the process of deep learning training, the phenomenon of overfitting often occurs when the scale of the training data are not large enough, and the feature of training data are over-repetitive learned. The overfitting phenomenon results in poor generalization ability of the model and poor prediction ability for new data. However, it usually requires a high cost to obtain a large amount of new data, so it is necessary to perform data enhancement on the collected raw data through algorithms. The keyframes of massage haptic data can be augmented by constructing a neural network based on the principle of a variational auto-encoder [29].

Figure 4 is a schematic diagram of a variational auto-encoder. The massage data samples {***X*_1_**,…,***X_n_***} are described by ***X***. In an ideal case, the distribution ***p*(*X*)** of ***X*** can be obtained according to {***X*_1_**,…,***X_n_***}, all massage data samples, including {***X*_1_**,…,***X_n_***} can be collected from ***p*(*X*)**. However, it is impossible to achieve in practice, so an additional variable z is introduced, and it is assumed that the posterior distribution ***p*(*Z|X*)** is a standard normal distribution. For any given sample ***X_k_***, it is assumed that there is a distribution ***p*(*Z|X_k_*)**, transform the distribution ***p*(*X*)** of the original massage tactile data into a distribution of ***x*** generated by ***z***: $p\left(x\right)=\sum_{z}p\left(X|Z\right)\cdot p\left(Z\right)$, so that ***Z_k_*** can be sampled from the distribution of ***z*** and reverted to ***X_k_***. The two parameters μ and $\sigma^{2}$ of the normal distribution ***p*(*Z|X*)** can be obtained by establishing a neural network fitting. The variational auto-encoder can be improved to a conditional variational auto-encoder by modifying the loss function of the neural network that fits the parameters of the normal distribution so that the mean of the normal distribution is close to the given condition. The key frames of the massage map group extracted by the frame difference method are sent to the conditional variational auto-encoder, and the original massage data can be greatly expanded to create new massage tactile data that are close to the original data by sampling and reverting the latent variables.

Each frame of massage map of each group is taken as the original sample of the variational auto-encoder and sent to the variational autoencoder neural network for training, and the network outputs new extended images. The original 100 sets of massage data for each type of massage technique are expanded into 900 sets of massage data. The original data and expanded data of the first type of massage technique are shown in Figure 5.

## 3. Neural Network Structure

Figure 6 is the network structure diagram, and Figure 7 is the algorithm flow chart of the network. The ***k*** in Figure 7 indicates the trained epoch. According to the sequential massage maps collected by the massage glove, this paper combines a 2D convolutional neural network and recurrent neural network to design a method that can not only extract the spatial domain features of a single massage map but also extract timing domain features between each map.

First, the preprocessed massage data are used as the input of the neural network, and the number of iterations of the network is set. The 2D convolutional neural network is used to extract the spatial domain features of the massage data, and the long short-term memory recurrent neural network is used to extract the time domain features of the massage data. The frame-attention mechanism is introduced between the two neural networks. The 2D convolutional neural network adopts the ResNet152 model pre-trained on the dataset ILSVRC-2012-CLS. Some weight parameters in the pre-trained ResNet-152 model are frozen to reduce the computational cost of training. The 2D convolutional neural network reduces the dimensionality of the massage map group corresponding to about 20 frames of each massage manipulation sample, and extracts features for encoding. Then, the encoded massage map group is arranged in chronological order, and the data dimension encoded by the 2D convolutional neural network becomes: (batch size, frames, CNN embed dim); the encoded data are pooled globally. After the frame attention block, the frame dimension of the massage map group obtain a weight value in the range of 0–1. At this time, the data dimension is: (batch size, frames, CNN embed dim); input the data into the Long Short-Term Memory (LSTM) recurrent neural network in the order of the frames dimension to train the timing domain features of the learning data, and add a linear layer after the recurrent neural network to reduce the data dimension to: (batch size, frames); After going through the softmax layer, the data output is normalized to 0–1, and the cross-entropy loss function is used to calculate the loss between the recognition result and the real label. The back-propagation algorithm updates the weight parameters of the network after each training epoch. After 50 rounds of training, a neural network that can recognize massage techniques is obtained.

### 3.1. Convolutional Neural Network

The convolutional neural network can process multi-dimensional data and achieve the purpose of data dimensionality reduction and feature extraction by performing sliding convolution operations on the previous layer of data. A convolutional neural network is mainly composed of an input layer, convolution layer, pooling layer, Relu layer, and fully connected layer. The convolutional layer can be seen as a set of filters that can learn parameters. Each filter is relatively small in space, but the depth is consistent with the input data. Filters are activated when the network sees certain types of visual features, and the specific feature can be the boundary in certain directions, or the spots of certain colors, etc. The role of the convolutional layer is to reduce the dimensionality of the data and extract features in the spatial domain. Usually, a pooling layer is periodically inserted between consecutive convolutional layers; its function is to gradually reduce the spatial size of the data, so that the number of parameters in the network can be reduced, and the computational resource consumption can be reduced. It can effectively control overfitting, and pooling can be divided into maximum pooling, average pooling, and L-2 pooling. The Relu layer introduces nonlinearity into the network and improves the generalization ability of the network. Usually, several linear layers are added at the end of the convolutional neural network to reduce the dimension of the data to the size that needs to be classified.

The main advantages of convolutional networks are [30]: (1) Sparse interaction, compared with fully connected matrix multiplication, the neurons in the layer are only connected to a small area in the previous layer, and the required weight parameters are greatly reduced, it can improve the overfitting phenomenon to a certain extent; (2) Parameter sharing, for the same convolution kernel, they share the same parameters, and can process all input data; (3) Equivariant representation, do translation in a small space for input data, it has less impact on the classification.

In this paper, the convolutional network module adopts the ResNet-152 model pre-trained on the dataset ILSVRC-2012-CLS, removes the last linear layer of the original ResNet-152 model, and adds two linear layers. The three dimensions—color channel, horizontal, and vertical pixels—are reduced to an embedding layer. The original input data dimensions of the neural network are: (batch size, frames, channels, image size ***x***, image size ***y***). After the data’s spatial features are extracted by the 2D convolutional neural network, the data’s dimension becomes: (batch size, frames, CNN embed dim). Some weight parameters in the pre-trained ResNet-152 model are frozen during network training to reduce the computational cost of training.

### 3.2. Attention Mechanism

A two-dimensional image has three dimensions: length, width, and channel. The channel attention module can automatically obtain the weight of the image channel dimension through network training. The channel attention mechanism model is shown in Figure 8. ***X*** and $\tilde{X}$ are input and output variables, respectively. ***C***, ***H***, and ***W*** represent the channel dimension, height dimension, and width dimension of variables, respectively. ***F_tr_*** represents any given transformation for variables, ***F_ex_*** represents excitation transformation for variables, ***F_scale_*** represents dimension reduction transformation for variables. After the convolution operation, the first step is to separate a bypass to perform a squeeze operation on the image and compress the length and width of the image separately into a real number, which is equivalent to a pooling operation with a global receptive field. The image channel dimensions remain unchanged. The second step is to perform the excitation operation on the squeezed image and generate weights for each channel through parameter w, which is learned to explicitly model the correlation between channels. Finally, the squeezed and excited data of the bypass are multiplied by the original convolved image data, and the weight is assigned to the image channel dimension.

The channel attention mechanism is improved to the frame attention mechanism for massage data, and the weights of the massage data in the frame dimension are assigned. The structure of the frame attention mechanism is shown in Figure 9. ***X*** and $\tilde{X}$ are input and output variables, ***r*** stands for linear layer dimensionality reduction coefficient. The dimension of the massage tactile data extracted from the spatial feature by the 2D convolutional neural network becomes: (batch size, frames, CNN embed dim), and the data dimension is transformed into: (batch size, CNN embed dim, frames) through dimension transposition, the data dimension is reduced to: (1, 1, frames) through a global pooling layer, and the data dimension is changed to: (1, 1, frames/***r***) through a linear layer, and a ReLU activation function is used to introduce nonlinearity to the network. After a linear layer, the data dimension is restored to (1, 1, frames), and then a layer of Sigmoid function is used to normalize the value of the frames between 0 and 1. At this time, the data dimension is: (1, 1, frames), multiply the data at this time to the original data extracted from the spatial feature by the 2D convolutional neural network, and the frame dimension of the original data is given a weight between 0–1. Finally, the data dimension is restored to (batch size, frames, CNN embed dim) through dimension transposition. Therefore, the channel attention mechanism is improved to the frame attention mechanism for tactile data, and the network can independently train the weights of massage data corresponding to different moments, which can enhance the network’s learning of data features and improve the network overfitting phenomenon.

### 3.3. Recurrent Neural Network

The recurrent neural network obtains the timing domain dependencies between sequence data by expanding the computational graph in the time domain. During the expansion process, the data corresponding to different moments passes through the same RNN computing unit, and these units share weights so that the network learns the contextual relevance. However, due to the weight-sharing characteristics of RNN units, they have serious problems with gradient explosion and gradient disappearance. To solve this problem, the literature [31] proposed a gated unit-based long-short-term memory neural network that maintains a cell state and controls the forgetting, increasing, and outputting of information by using input gates, forgetting gates, and output gates. The derivative calculation in the traditional RNN multiplication form is changed into an accumulation form, thereby avoiding the problem of gradient disappearance, and LSTM can process longer time series data.

As shown in Figure 10, it is assumed that the features extracted by the convolutional network corresponding to a group of massage techniques are: $X=\left(x_{1},x_{2},\ldots ,x_{T}\right)\in R^{m},i=1,2,\ldots ,T$, the vectors in *X* are sequentially input into the LSTM network, and for the input xt at time t, the calculation process through each gate is as follows:(1)Input gate:
(2)$$i_{t}=\sigma \left(W_{i}\cdot \left[h_{t-1},x_{t}\right]+b_{i}\right)$$(2)Forgetting gate:
(3)$$\;f_{t}=\sigma \left(W_{f}\cdot \left[h_{t-1},x_{t}\right]+b_{f}\right)$$
(4)$$C_{t}=f_{t}\cdot C_{t-1}+i_{t}\cdot \hat{C_{t}}$$
(5)$$\hat{C_{t}}=\tanh\left(W_{C}\cdot \left[h_{t-1},x_{t}\right]+b_{c}\right)$$(3)Output gate:
(6)$$o_{t}=\sigma \left(W_{o}\cdot \left[h_{t-1},x_{t}\right]+b_{o}\right)$$
(7)$$h_{t}=o_{t}\cdot \tanh\left(C_{t}\right)$$
where ***h_t_*_−1_** represents the memory output from the gating unit at the previous moment, ***x_t_*** represents the feature input at the current moment, ***i_t_*** represents the input gate value, ***f_t_*** represents the forgetting gate value, ***o_t_*** represents the output gate value, ***C_t_*** represents the current cell state, ***W_i_*** and ***W_C_*** are the input gate connection weights of the LSTM network, ***b_i_*** and ***b_c_*** are their biases, and ***W_f_*** and ***b_f_*** are connection weights and biases of the forget gate of LSTM network, ***W_o_*** and ***b_o_*** are connection weights and biases of the output gate of LSTM network, ***σ(.)*** is the neural network sigmoid activation function.

The data whose frame dimension has been given weights (batch size, frames, CNN embed dim) input into the long short-term memory recurrent neural network to train the timing domain features of the data and connect the output hidden layer ***h_n_*** corresponding to the last time series of the long short-term memory recurrent neural network to a linear layer to reduce the data dimension to (batch size, ***N*** Categories). The loss of the network adopts the cross entropy loss function, the network optimizer adopts Adam optimizer, and the network is trained to realize the purpose of identifying massage techniques.

## 4. Experiment and Results

In the experiment of this paper, the dataset generated by the variational auto-encoder is divided into a training set and a dataset according to the ratio of 3:1. There are three types of massage techniques: each type of massage technique contains 900 samples, and each sample contains 20 frames of massage maps in chronological order.

To verify the performance of the RestNet152+RNN neural network structure built in this paper, we also used two network structures, AlexNet+RNN [32] and 3DCNN [23], to train the massage dataset. The parameter settings of the three network structures are shown in Table 1. The learning rate, batch size, dropout probability, training epochs, optimizer, and loss function of the three network structures are set to be the same. Compared with the structures of ResNet152+RNN and AlexNet+RNN, the convolution modules of the two networks are different. The difference is that the former convolution module adopts the ResNet152 structure, including 151 convolution layers with a deep convolution depth, and the latter convolution module adopts the AlexNet structure, including 5 layers of convolution layers and 3 maximum pooling layers. The 3DCNN network structure also realizes the extraction of tactile image timing information by adding continuous frames of tactile images as depth channels of the convolution kernel. The curve of the recognition accuracy of the test set with the training epochs of the three networks is shown in Figure 11.

As shown in Figure 11, under the condition that the learning rate, batch size and other parameters are the same, the network recognition accuracy of the ResNet152+RNN structure converges to 100%; the recognition accuracy of the AlexNet+RNN network structure is stable at 66% under 50 training cycles; The accuracy of the 3DCNN network structure oscillates around 70% under 50 training cycles. The confusion matrix of the three networks is shown in Figure 12. From the confusion matrix, the ResNet152+RNN network can identify the three massage actions well. The AlexNet+RNN network and 3DCNN network have high recognition accuracy for the third massage action, but both have poor recognition ability for the first and second massage actions. The two kinds of actions will be confused during the recognition process, especially when the first kind of action is easily misidentified as the second kind of action. This should be because the first and second massage actions are both rubbing actions; only the directions are different. When a variational autoencoder is used to expand the dataset, some noise may be introduced into the massage dataset, making it difficult to distinguish between the two massage actions. The ResNet152 module with a deeper network structure can better extract the deep features of massage data as a convolutional layer, so we choose the ResNet152 module as the convolutional module of our network.

Since the self-made massage dataset has few data categories, the data complexity is low, and additional noise will be introduced when the variational auto-encoder expands the dataset, and the designed neural network model has a complex structure and strong learning ability, it is easy to mistake the error of the data in the training set as the general law of the data itself, resulting in the phenomenon of overfitting and a decrease in the generalization ability of the model. Therefore, it is necessary to set the dropout probability of parameters in the training process to weaken the network’s learning of data errors, or introduce the attention mechanism into the model to assign weights to each frame of massage map to enhance the network’s extraction of data features.

Based on the ResNet152+RNN network structure, four network structures are constructed: the dropout probability is 0.3, and the attention mechanism is not introduced; the dropout probability is 0.3, and the attention mechanism is introduced; the dropout probability is 0, and the attention mechanism is not introduced; the dropout probability of the parameters is 0, and the attention mechanism is introduced. Four kinds of network structures are used to train the massage dataset, respectively. The curve of the recognition accuracy of the test set with the training epochs is shown in Figure 13.

When the dropout probability of the model parameters is 0.3 and the attention mechanism is not introduced, the recognition accuracy of the massage test set converges in the 17th training cycle. Compared with the other three models, its convergence speed is slow and the overfitting phenomenon appears in the 16th training cycle; when the dropout probability of model parameters is 0.3 and the attention mechanism is introduced, the recognition accuracy of the massage test set does not converge, and the overfitting phenomenon is serious, indicating that the dropout probability of parameters may lead to the failure of learning some attention parameters, and some features of the data itself are not available, which makes the model structure worse; when the dropout probability of model parameters is 0 and the attention mechanism is not introduced, the recognition accuracy of the massage test set does not converge, and the recognition accuracy oscillates between 95% and 100%, the phenomenon of overfitting occurs; when the dropout probability of model parameters is 0 and the attention mechanism is introduced, the recognition accuracy of the massage test set converges in the 8th training cycle, the convergence speed is rapid, and there is no overfitting phenomenon during the training cycle, indicating that adding the frame attention mechanism can improve the model structure to some extent, enhance the extraction of data features, speed up the model convergence speed, and effectively improve the network overfitting phenomenon.

This paper uses three indicators to evaluate the four network structures: recognition accuracy, recall, and convergence epoch. Recognition accuracy indicates the proportion of all correctly identified samples in the massage dataset to the total number of samples; recall indicates the proportion of correctly identified samples in a certain type of massage technique to the total number of samples of this type. The lowest recall of the three action categories is selected here; the convergence epoch represents the epoch in which the recognition accuracy is stable and does not decrease. High recognition accuracy and recall, as well as a short convergence epoch, indicate excellent model performance. The models of the last three training epochs of the four network structures were used to recognize the entire dataset, and the average recognition accuracy and recall was taken. Table 2 compares the three performance indicators of the four network structures. It can be obtained from the table that the network structure with the dropout probability of model parameters of 0 and introduction of the attention mechanism has the highest recognition accuracy and recall and the shortest convergence epoch. Compared with the network structure with dropout probability of model parameters of 0.3 and introduction of attention mechanism and the network structure with dropout probability of model parameters of 0 and no introduction of attention mechanism, the recognition accuracy is increased by 0.74% and 1.61%, respectively, and recall increased by 2.29% and 4.82%, respectively. Compared with the network structure with dropout probability of model parameters of 0.3 and no introduction of attention mechanism, its convergence epoch is 12 training epochs earlier. Therefore, the introduction of the attention mechanism can effectively improve the network structure, reduce the epoch required for network convergence, and improve the overfitting phenomenon to enhance the generalization ability of the network.

## 5. Conclusions

A method combining a convolutional neural network, long-term neural network, and attention mechanism is proposed to identify the massage techniques in this paper. The massage maps of masseur are collected through a self-made massage glove and its data acquisition system; the frame difference method and the conditional variational auto-encoder were used for data processing to make a massage dataset; by combining the convolutional neural network and the recurrent neural network, it realizes extracting features of massage maps in spatial and timing domain, and improves the channel attention mechanism into the frame attention mechanism for tactile images and introduces it into the network structure, so that the weight of each massage maps in the frame dimension is automatically trained and learnt to enhance the extraction of data features. By changing the dropout probability of model parameters during training and whether to introduce an attention mechanism, four network structures based on the ResNet152+RNN network structure are constructed to learn the dataset. Experiments show that compared with the other three network structures, the network structure with the dropout probability of model parameters of 0 and the introduction of the attention mechanism has the highest recognition accuracy and recall, reaching 100% in the self-made massage dataset. The convergence epoch is short, only 5 cycles are required, and there is no overfitting phenomenon. The proposed network structure can learn the features of the massage map group in both the spatial domain and timing domain and can effectively improve the overfitting phenomenon of the network when the dataset complexity is low. We also compare the ResNet152+RNN network structure with the AlexNet+RNN network structure and the 3DCNN network structure. The results show that the ResNet152+RNN network structure performs better in massage action recognition tasks. Our method has a good application in identifying massage techniques and provides a solid foundation for guiding robotic massage applications.

In future work, in the aspect of sensor and its data acquisition system: it can be considered to improve the flexibility and robustness of the sensor from the sensing principle, increase the speed of the data acquisition circuit and improve the crosstalk phenomenon in the data acquisition process; in the aspect of neural network: consider to improve the identification of only a single massage technique to segment and identify a single massage technique from a combination of massage techniques.

## Figures and Tables

**Figure 1 sensors-22-05632-f001:**
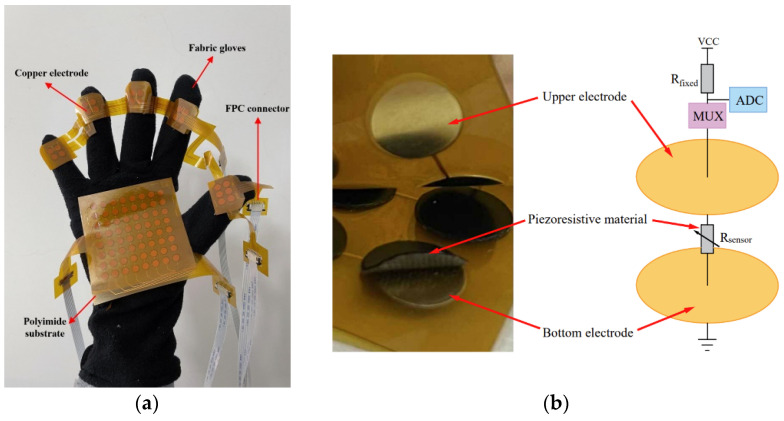
Physical diagram and schematic diagram of the tactile sensor. (**a**) Physical image of the tactile sensor integrated into the glove; (**b**) Schematic of the tactile sensor.

**Figure 2 sensors-22-05632-f002:**
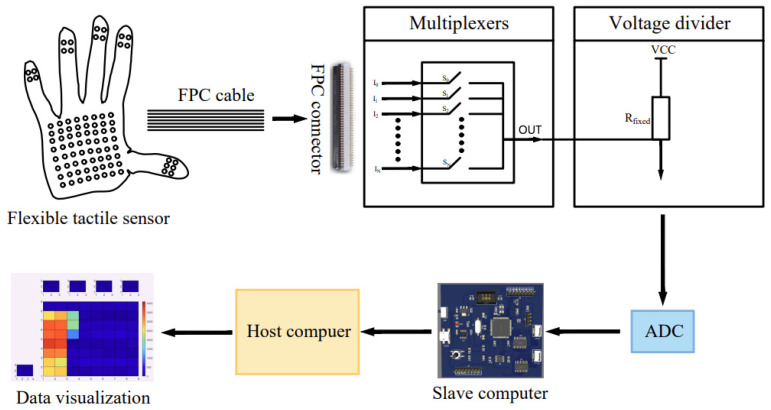
Diagram of the massage data acquisition system.

**Figure 3 sensors-22-05632-f003:**
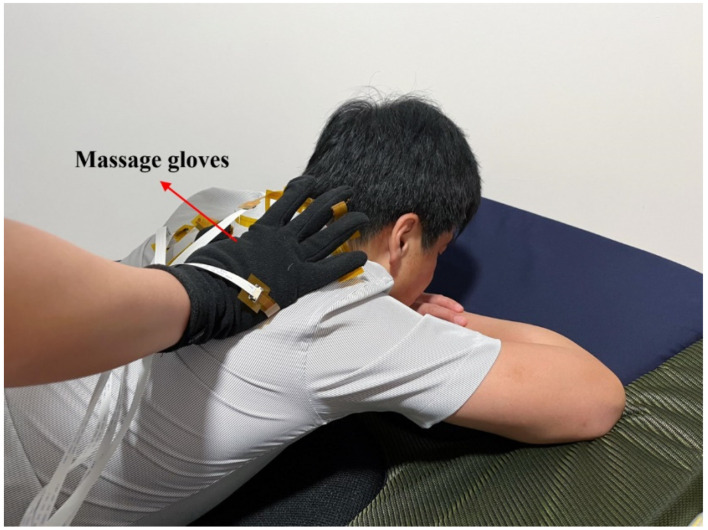
Diagram of the massage data collection process.

**Figure 4 sensors-22-05632-f004:**
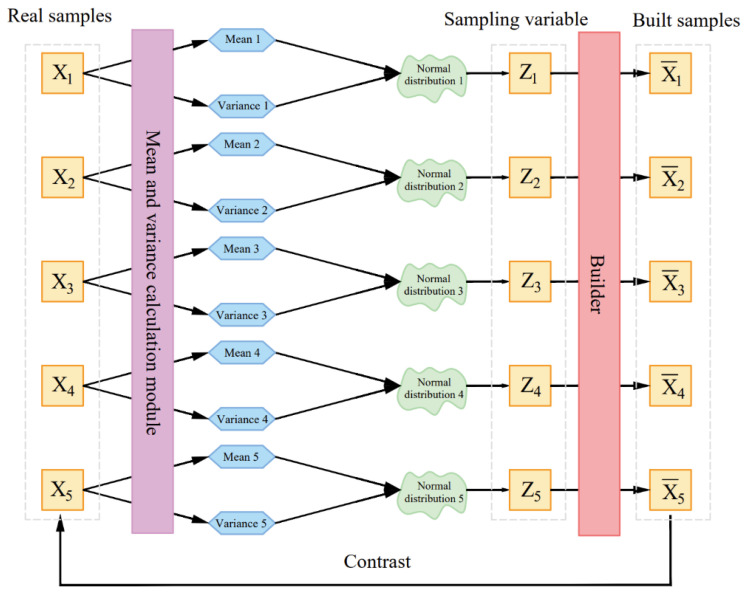
Schematic diagram of a variational auto-encoder.

**Figure 5 sensors-22-05632-f005:**
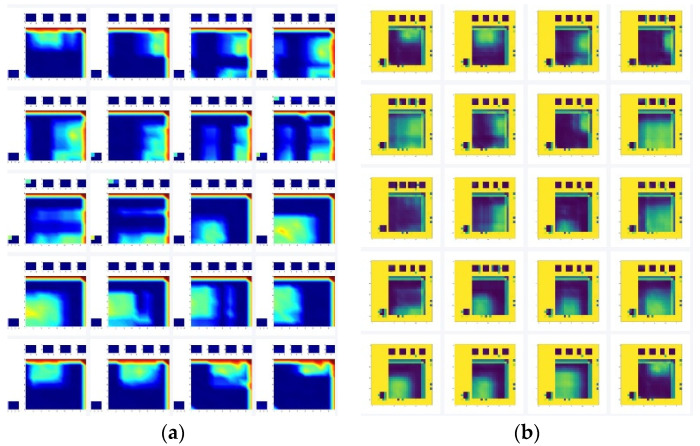
Effect diagram of the variational auto-encoder. (**a**) Origin data; (**b**) Expanded data.

**Figure 6 sensors-22-05632-f006:**
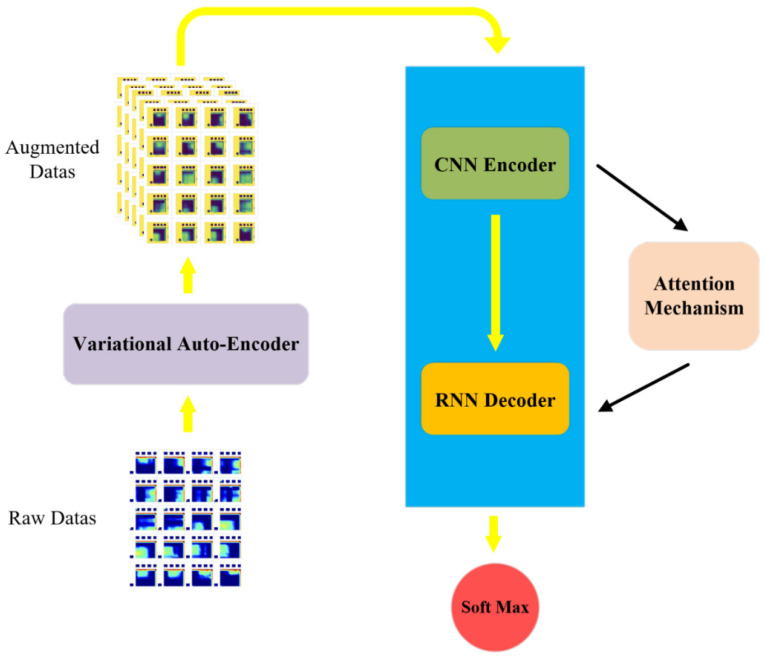
Network structure diagram.

**Figure 7 sensors-22-05632-f007:**
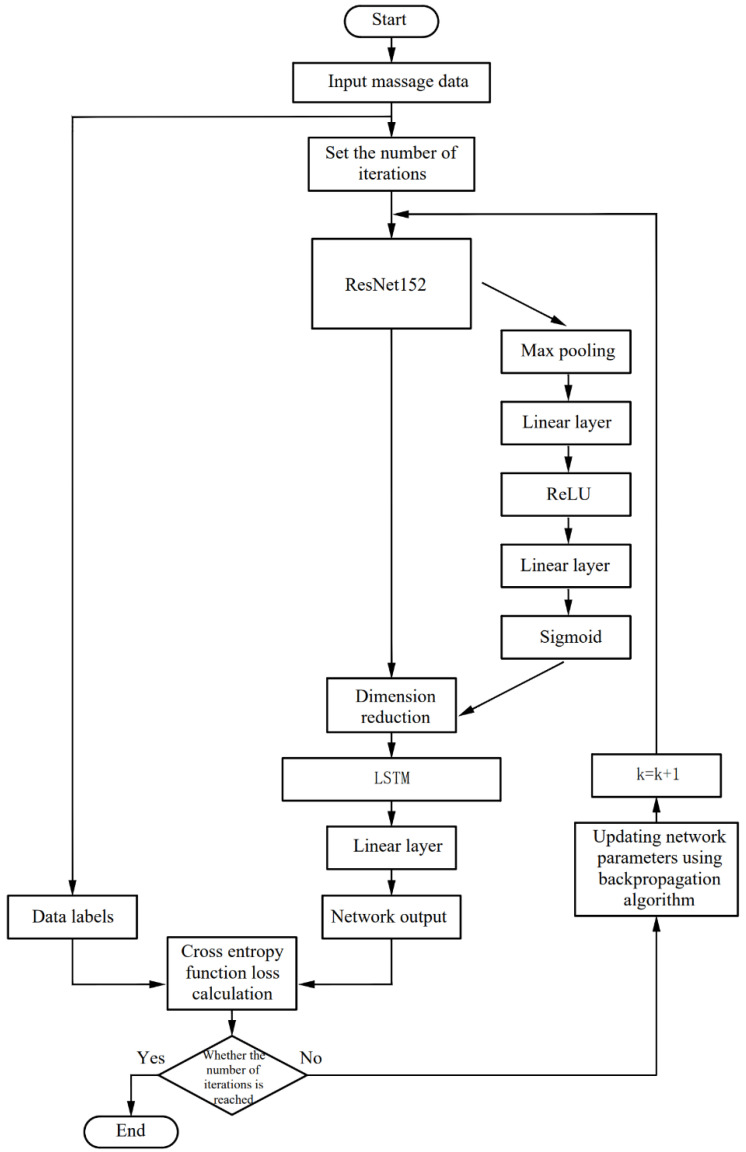
Algorithm flow chart of the network.

**Figure 8 sensors-22-05632-f008:**
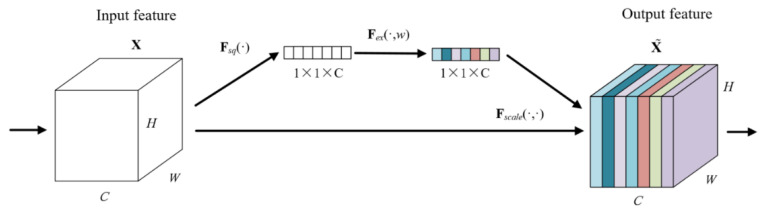
Channel attention mechanism model.

**Figure 9 sensors-22-05632-f009:**
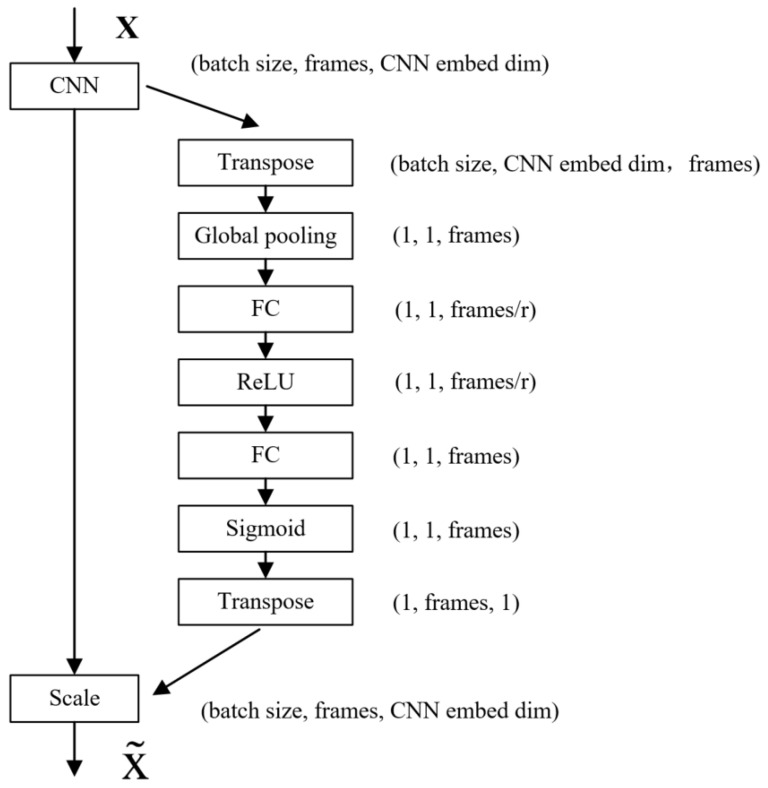
Frame attention mechanism structure.

**Figure 10 sensors-22-05632-f010:**
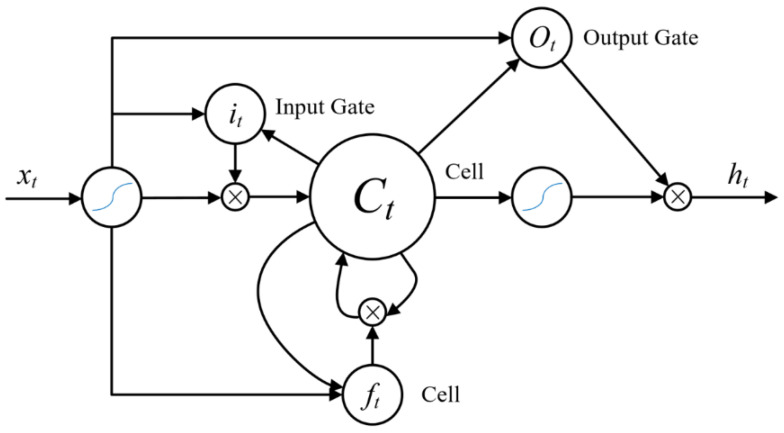
Long short-term memory neural network model.

**Figure 11 sensors-22-05632-f011:**
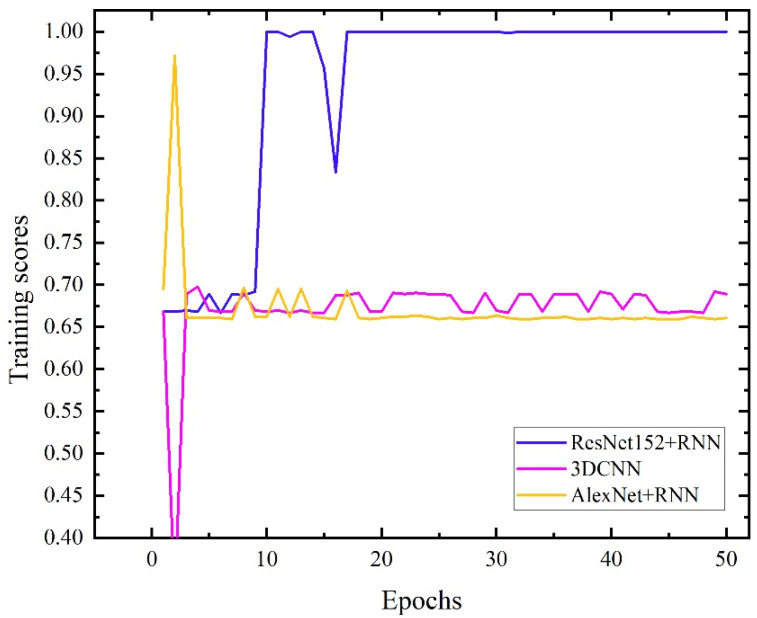
Test data training score curves of the three networks.

**Figure 12 sensors-22-05632-f012:**
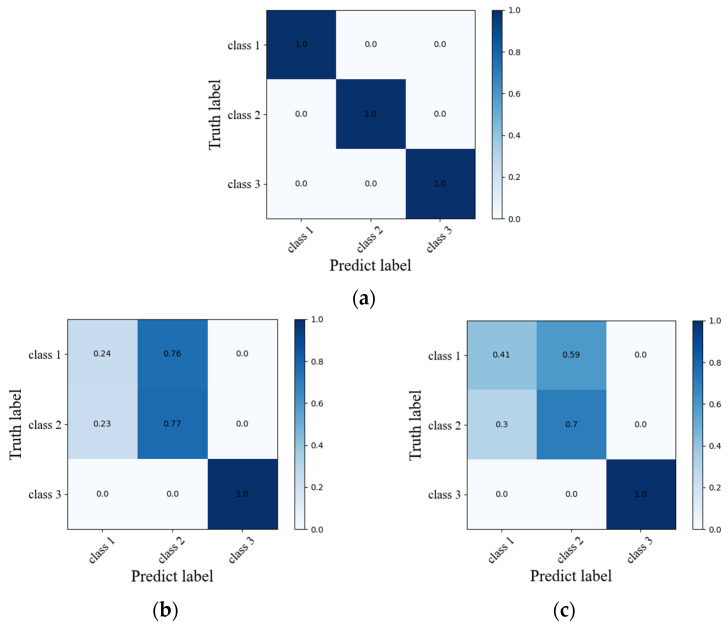
Confusion matrix of the three networks. (**a**) ResNet152+RNN; (**b**) AlexNet+RNN; (**c**) 3DCNN.

**Figure 13 sensors-22-05632-f013:**
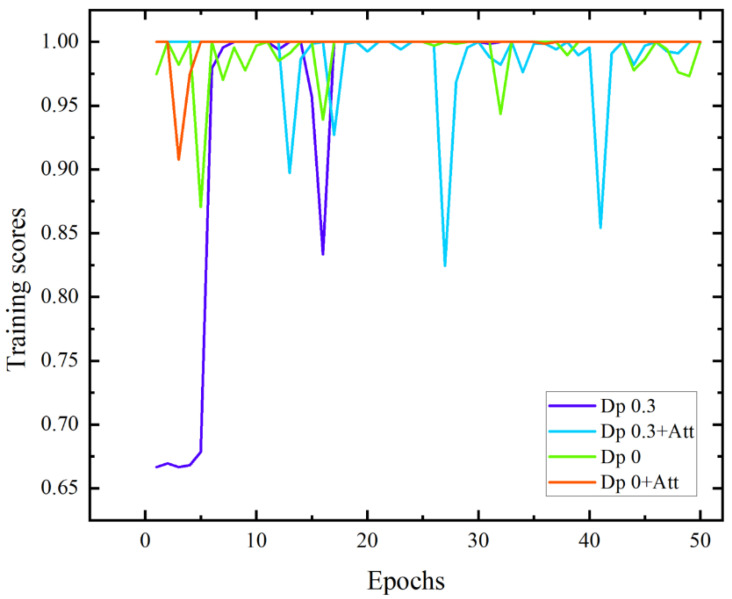
Test data training score curves of the four models.

**Table 1 sensors-22-05632-t001:** Parameter settings of the three networks.

Parameter	Network Structure		
	ResNet152+RNN	AlexNet+RNN	3DCNN
Convolutional layers	151	5	2
Linear layers	5	5	3
Learning rate	1 × 10^−3^	1 × 10^−3^	1 × 10^−3^
Batch size	32	32	32
Dropout probability	0.3	0.3	0.3
Training epochs	50	50	50
Optimizer	Adam	Adam	Adam
Loss function	Cross Entropy	Cross Entropy	Cross Entropy

**Table 2 sensors-22-05632-t002:** Comparison of performance indicators of four network structures (%).

Item	Dp 0.3	Dp 0.3+Att	Dp 0	Dp 0+Att
Accuracy	100	99.26	98.39	100
Recall	100	97.71	95.18	100
Epoch	17	—	—	—

## Data Availability

The data that support the findings of this study are available from the corresponding author upon reasonable request.

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
