# Peer review of "Recognition Method of Massage Techniques Based on Attention Mechanism and Convolutional Long Short-Term Memory Neural Network"

_sensors, 2022, doi:10.3390/s22155632_

Round 1

Reviewer 1 Report

1. This is a simple logic circuit system. Can the MCU handle a large number of neural-like CNN operations?

2. Why not use FPGA design, but use complex artificial intelligence computing, please explain the reasons.

3. Compare with the traditional method.

4. Since CNN is used, please compare the performance with AlexNet.

5. Mostly are photos or flowcharts. A complete design concept diagram and algorithm flow chart are missing. The result map is only Figure 10, which is very weak. Please strengthen the presentation of the results!

Reviewer 2 Report

This paper utilizes flexible tactile sensors to acquire data from various touch points and the neural network is developed to realize a massage map with a time domain. This paper could be accepted after addressing a few typos in the paper, for example: (page 3) the contribution 2 and 3 mentioned on page 3 are same; (page 4) the representation of the sections at the end of the introduction is not matching with the respective sections. 

Reviewer 3 Report

This manuscript proposes a new deep learning strategy for Identifying the massage techniques. The approach combines a convolutional neural network, a long-term neural network, and an attention mechanism to recognize multiple consecutive massage maps.

The manuscript is well structured and written. However, there are some concerns that the authors need to fix.

1.       Figure 6 needs more explanation and clarification.

2.       Insert a table with all details about the CNN and RNN architecture, such as the number of layers, the kernel size per layer, and others. “How can anyone reproduce the work if the details are not presented.”

3.       There are several variables in Figure 7, but they are not in the text. All variables need to be explained.

4.       Same situation for Figure 8 and Figure 9

5.       The equations 2 to 8 are lost. Why are these equations presented?

6.       The analysis of the results is very weak and poor. It is mandatory the authors improve it.

7.       There is similar work in the literature? If yes, make a comparison. 

Round 2

Reviewer 1 Report

The author's revision has been completed and is recommended for publication.

Reviewer 3 Report

The authors made all the requested suggestions.